# Utilizing CNNs for classification and uncertainty quantification for 15 families of European fly pollinators

Thomas Stark[1]*, Michael Wurm[1], Valentin Stefan[2,3,4], Felicitas Wolf[2,3,4], Hannes Taubenböck[1,5], Tiffany M. Knight[2,3,4]

1 German Aerospace Center (DLR), German Remote Sensing Data Center (DFD), Oberpfaffenhofen, Germany, 2 Department of Community Ecology, Helmholtz Centre for Environmental Research - UFZ, Halle (Saale), Germany, 3 German Centre for Integrative Biodiversity Research (iDiv) Halle-Jena-Leipzig, Leipzig, Germany, 4 Institute of Biology, Martin Luther University Halle-Wittenberg, Halle (Saale), Germany, 5 Institute of Geography and Geology, University of Wuerzburg, Wuerzburg, Germany

* thomas.stark@dlr.de

**Data availability statement:** All data is available and mentioned in the section Data and code availability or directly at https://github.com/stark-t/PAI_diptera/tree/main/data/PAI_Diptera_family_GBIF.

## Abstract

Pollination is essential for maintaining biodiversity and ensuring food security, and in Europe it is primarily mediated by four insect orders (Coleoptera, Diptera, Hymenoptera, Lepidoptera). However, traditional monitoring methods are costly and time consuming. Although recent automation efforts have focused on butterflies and bees, flies, a diverse and ecologically important group of pollinators, have received comparatively little attention, likely due to the challenges posed by their subtle morphological differences. In this study, we investigate the application of Convolutional Neural Networks (CNNs) for classifying 15 European pollinating fly families and quantifying the associated classification uncertainty. In curating our dataset, we ensured that the images of Diptera captured diverse visual characteristics relevant for classification, including wing morphology and general body habitus. We evaluated the performance of three CNNs, ResNet18, MobileNetV3, and EfficientNetB4 and estimated the prediction confidence using Monte Carlo methods, combining test-time augmentation and dropout to approximate both aleatoric and epistemic uncertainty. We demonstrate the effectiveness of these models in accurately distinguishing fly families. We achieved an overall accuracy of up to 95.61%, with a mean relative increase in accuracy of 5.58% when comparing uncropped to cropped images. Furthermore, cropping images to the Diptera bounding boxes not only improved classification performance across all models but also increased mean prediction confidence by 8.56%, effectively reducing misclassifications among families. This approach represents a significant advance in automated pollinator monitoring and has promising implications for both scientific research and practical applications.

## Introduction

Globally, insects are the most important animal pollinators, and therefore monitoring and understanding insect pollination is critical to biodiversity conservation and food security

**Funding:** The author(s) received no specific funding for this work.

**Competing interests:** The authors have declared that no competing interests exist.

[1–3]. Current insect monitoring methods are expensive, labor-intensive, and slow, underscoring the need for efficient and automated approaches. Recent automation efforts have predominantly targeted butterflies and bees [4,5]. However, fly pollinators provide substantial pollination services to wild and crop plants [6]. In Europe, 15 fly families are known to contribute to pollination, representing over 5000 species [7]. Classifying flies from images poses challenges due to morphological similarities across some families [8]. The limited availability of images for flies, compared to more charismatic taxa of pollinators, makes it difficult to create a comprehensive dataset, which is essential for successfully training Convolutional Neural Networks (CNNs) [9,10]. CNNs are a class of deep learning models particularly well-suited for image recognition tasks due to their ability to automatically learn hierarchical spatial features from raw pixels [11,12].

Recent years have witnessed the widespread adoption of deep learning models for image classification, significantly advancing various fields, such as medicine [13], urban studies [14–16], and agricultural sciences [17,18]. Furthermore, deep learning methods have increasingly become a key tool in the field of ecology and biodiversity, offering new avenues for research and conservation.

One of the most significant contributions of deep learning is in species identification and monitoring. The development of deep learning tools for the classification of arthropods initially focused on specific taxa (e.g., 16 species of mosquitoes) and the identification of museum species with uniform image backgrounds [19]. For example, images of arthropod wings taken in controlled settings, such as under a microscope, have been used to identify various groups of bees [20], butterflies [21,22], and syrphid flies [23]. The use of CNNs for the identification of arthropods has expanded from a few taxa to multiple taxa, utilizing an increasing number of images. Examples include the identification of nine genera of tiger beetles with 380 images [24], eight groups of arthropods with nearly 20,000 images [25], and 36 species of bumblebee with nearly 90,000 images [26].

However, pollinating flies have not yet been systematically addressed in this line of research, despite being the world's second most important and abundant pollinators after bees [6,27]. Flies, particularly syrphids and other Diptera, make vital contributions to both natural ecosystems and agricultural systems, especially in high-latitude or high-altitude environments where bee populations decline or shift [28,29]. Long-term studies have shown shifts in pollinator communities over time, with flies playing an increasingly central role in many regions [29]. Given their ecological importance and diversity, it is crucial that future deep learning models are developed to reliably identify pollinating flies, at least to the family level, to ensure more comprehensive monitoring, support biodiversity assessments, and inform conservation strategies.

Using image classification techniques for pollinator identification often lacks insight into the confidence levels in its predictions. This is particularly relevant in challenging cases where image quality or species similarity can impact accuracy [19,30]. Ensuring trustworthiness is critical and uncertainties play a crucial role in this case. This study delves into integrating uncertainty estimation methods within deep learning models to enhance predictive trustworthiness by explicitly addressing aleatoric and epistemic uncertainties. We further clarify how these uncertainty estimates compare to previous approaches in ecological deep learning studies and demonstrate their specific value for insect classification, where limited training data and ambiguous visual cues are common [31,32].

The uncertainty estimation goes beyond traditional predictions, providing insight into the confidence and reliability of the model [33]. Incorporating these uncertainty estimation methods is crucial in real-world applications where decision-making about pollination abundance

counts is based on accurate and reliable predictions. By transparently quantifying uncertainties, the models not only improve interpretability, but also provide a measure of confidence in their predictions [34]. This study outlines the methodologies used to capture and quantify uncertainties, contributing to the broader goal of establishing trust in deep learning models for image classification. Specifically, we use a combination of Test-Time Augmentation (TTA) and Test-Time Dropout (TTD), leveraging Monte Carlo sampling to approximate predictive uncertainty, as proposed by Gal and Ghahramani [35]. One key research question is whether CNNs can effectively differentiate between the 15 families of pollinating flies, including a comparison of the impact of provided images versus cropping on classification accuracy.

This research focuses on the classification of a diverse range of Diptera families, employing a methodology that emphasizes both image refinement and robust confidence estimation. A key aspect involves comparing the performance of classification models when working with two distinct image datasets: The first dataset retains the original images sourced from the Global Biodiversity Information Facility (GBIF), while the second dataset relies on the same images but cropped to expert-defined bounding boxes that focus on the full specimen. To thoroughly evaluate these strategies, the approach compares three state-of-the-art CNN architectures ResNet18, MobileNetV3, and EfficientNetB4 across both image sets. Using a focus on the targeted areas, this approach aims to highlight defining features and reduce visual noise, ultimately guiding the classifier to more pertinent details.

To ensure that the classification framework remains transparent and interpretable, the research integrates uncertainty quantification methods. These techniques generate meaningful confidence values that reflect the level of certainty of the model, allowing researchers to better understand the quality and reliability of the predictions [36–38]. Such insights are particularly important for closely related Diptera families, where subtle differences in morphology may lead to misclassifications [8]. By examining groups with overlapping characteristics, specifically families such as Fannidae, Muscidae, and Tachinidae, the research attempts to refine classification boundaries and improve discriminative abilities.

## Materials and methods

### Describing 15 Diptera families

This research focuses on a set of 15 fly families selected not only for their distinctive morphological and phylogenetic traits, but also because they are known to provide pollination services throughout Europe [7]. A visual example fo each of the 15 families can be seen in Fig 1. These particular families embody a diverse array of species richness and ecological functions, providing a testbed for exploring classification challenges. The variation in the number of species across these families is notable; for example, while Sepsidae comprises around 48 known species in Europe, Tachinidae includes approximately 877 known species [7]. Such disparities in species richness can introduce significant variability in the visual characteristics of the specimens, influencing classification difficulty and the subsequent performance of CNN models.

The dataset includes fly families selected to represent a range of morphological differences and evolutionary relationships, providing visual characteristics relevant for training and validating the classification models. For each family the number of total images and the unique species and genus within each family within the dataset are seen in Table 1. Ultimately, understanding these nuances is critical to improving the accuracy, reliability, and interpretability of automated identification systems. Further morphological cues, particularly wing venation, provide crucial identification features in Diptera. For example, Syrphidae possess the vena

**Table 1. For each family the number of total images and the unique species and genus within each family within the dataset are shown.**

| Family | Images | Species | Genera |
|---|---|---|---|
| Anthomyiidae | 1488 | 132 | 31 |
| Bombyliidae | 2239 | 151 | 40 |
| Calliphoridae | 1840 | 37 | 13 |
| Conopidae | 2356 | 57 | 10 |
| Empididae | 1632 | 144 | 14 |
| Fanniidae | 521 | 36 | 2 |
| Hybotidae | 1739 | 93 | 17 |
| Muscidae | 2163 | 212 | 41 |
| Sarcophagidae | 1346 | 77 | 22 |
| Scathophagidae | 2044 | 64 | 24 |
| Sepsidae | 2363 | 32 | 10 |
| Stratiomyidae | 2538 | 115 | 29 |
| Syrphidae | 2321 | 485 | 82 |
| Tabanidae | 2360 | 97 | 12 |
| Tachinidae | 2325 | 370 | 190 |

spuria, a vein nearly exclusive to this family, while Empididae also display distinctive venation patterns. Calyptrate and acalyptrate flies may share similar wing venation, but calyptrates can be further distinguished by the presence of vibrissae bristles above the mouth. Additional traits such as body hair (e.g., hypopleural bristles), scutellum shape, and bristle positioning on the thorax and legs also aid in family-level classification. Despite the subtlety of these features, many experienced taxonomists can identify families such as Bombyliidae, Empididae, and Syrphidae by general habitus alone, even from limited image quality, a useful analogy for the capability of CNNs to learn from complex but non-explicit visual cues.

## Image acquisition

The images were obtained from GBIF with a focus on Europe, to maximize species richness. For families with abundant images, a random selection was applied to ensure a diverse dataset. Although it was not possible to automatically filter specifically for images of insects on flowers, efforts were made to remove images from museum collections, laboratory settings, fossils, duplicates, and other irrelevant sources or life stages. Due to varying levels of public interest, some species have a disproportionately large number of URLs available for image download, while others have significantly fewer resources. This results in a long-tail distribution of available images across the taxonomic spectrum of species and families. To address this imbalance, URLs were sampled as uniformly as possible across species within each family. Subsequently, a manual review was conducted to remove unwanted images, such as those depicting insect parts under magnification or misidentified specimens, thus ensuring a comprehensive and accurate dataset across species.

Expert-defined bounding boxes were created to enhance image accuracy by focusing on relevant features and reducing background noise. This approach is intended to improve the performance of CNN models in classifying the various families of pollinating flies.

## Data sampling

The dataset, comprising a total of 29,374 images, is partitioned into three distinct subsets, namely the training, validation, and testing datasets. To ensure a well-balanced and representative distribution across the 15 Diptera families, a stratified approach is employed. Each

family contributes 60% of its images to the training set, allowing the models to learn from a wide and diverse range of examples. Subsequently, 20% of the images from each family are assigned to both the testing and validation sets as seen in Table 2. This strategic distribution guarantees that each Diptera family is adequately represented in all datasets, thus creating a comprehensive understanding of the nuances within the characteristics of each family. This data partitioning forms a critical aspect of model training and evaluation, facilitating robust and reliable performance assessments across diverse taxonomic groups.

## Selection of CNNs

In this study, we used three distinct CNNs to compare their performance on our dataset. MobileNetV3 Large, ResNet18, and EfficientNetB4. Each network was chosen based on specific attributes that align with our research goals.

1. MobileNetV3 Large [39] is designed to be both fast and efficient, making it particularly suitable for deployment in environments with limited computational resources. This model has approximately 5.4 million parameters, striking a balance between performance and efficiency. We selected MobileNetV3 Large to deliver high-speed inference without compromising accuracy significantly.
2. ResNet18 [41], a member of the Residual Networks family, is well-regarded for its robust performance across various scientific fields. With approximately 11.7 million parameters, ResNet18 is relatively lightweight, yet highly effective, thanks to its residual learning framework that eases the training of deep networks. This model was chosen for its proven efficacy in handling complex and diverse datasets.
3. EfficientNetB4 [42] is the largest of the three models in terms of parameters, with approximately 19 million parameters. It uses a compound scaling method that uniformly scales network dimensions, resulting in improved performance without the corresponding increase in computational cost. EfficientNetB4 is used to deliver high accuracy due to its larger size and advanced architectural design.

**Table 2**. **Number of images in the Training (60%), Validation (20%), and Testing (20%) dataset.**

| Family | Training | Validation | Testing |
|---|---|---|---|
| Anthomyiidae | 892 | 298 | 298 |
| Bombyliidae | 1343 | 448 | 448 |
| Calliphoridae | 1104 | 368 | 368 |
| Conopidae | 1413 | 471 | 472 |
| Empididae | 979 | 326 | 327 |
| Fanniidae | 312 | 104 | 105 |
| Hybotidae | 1043 | 348 | 348 |
| Muscidae | 1297 | 433 | 433 |
| Sarcophagidae | 807 | 269 | 270 |
| Scathophagidae | 1226 | 409 | 409 |
| Sepsidae | 1417 | 473 | 473 |
| Stratiomyidae | 1522 | 508 | 508 |
| Syrphidae | 1392 | 464 | 465 |
| Tabanidae | 1416 | 472 | 472 |
| Tachinidae | 1395 | 465 | 465 |
| **Total** | **17558** | **5856** | **5861** |

## Uncertainty quantification

Understanding and quantifying uncertainties in predictions are crucial for robust model deployment. We addressed two distinct forms of uncertainty, aleatoric and epistemic, through the application of Monte Carlo uncertainty approximations, whereby multiple classification outputs are averaged to derive confidence values.

Aleatoric uncertainty arises from inherent variability and randomness within the data itself [36]. Our model was designed to capture and quantify this type of uncertainty using test-time augmentations (TTA). This allows us to account for situations where the input data exhibit ambiguity or contain inherent noise. Aleatoric uncertainty might manifest itself in scenarios with subtle or ambiguous visual features.

We provide a random set of image augmentations during both the training phase and when the model is applied to the test dataset. This TTA approach ensures that the model encounters a diverse range of augmented inputs during training and testing, facilitating its ability to generalize and accurately assess uncertainty in real-world scenarios [43,44]. The augmentations were grouped into four categories as presented in Table 3: (i) basic augmentations (horizontal flip and 90° rotation, both with (p=1.00); (ii) color augmentations (jitter, channel shuffle, Gaussian noise, blur, and sharpness, each with (p=0.1); (iii) geometric distortions (thin plate spline, random cropping, and erasing, each with (p=0.1); and (iv) mixing-based augmentation using CutMixV2 (p=0.1). All augmentations were applied stochastically across the ensemble predictions and averaged to obtain final probabilities, following established practices for test-time uncertainty estimation [43].

Epistemic uncertainty, on the other hand, is rooted in the limitations of the model's knowledge [36]. It reflects uncertainty arising from a lack of understanding or exposure to various data during training. We addressed epistemic uncertainty by incorporating test-time dropout (TTD). This helps the model recognize when faced with unfamiliar patterns not encountered during training, reducing the uncertainty associated with knowledge gaps [45]. For the dropout method we use the same value of 0.3 as shown in [14].

Expanding on using Monte Carlo samples generated by TTD for uncertainty estimation [14,43], our method extends the uncertainty estimation technique. We utilize Monte Carlo samples from TTA as well as from TTD. Our goal is to characterize the distribution, specifically the predictive posterior distribution of $\bar{y}$. This is achieved by training the neural network as if it were a standard network, incorporating dropout layers after each layer with weight parameters and conducting $T$ predictions.

**Table 3. Overview of data augmentation techniques used during test-time augmentation (TTA) and the percentage value if it is applied (p).**

| Group | Techniques | p |
|---|---|---|
| Basic | Horizontal Flip | 1.0 |
| | Rotation 90° | 1.0 |
| Color | Color Jitter | 0.1 |
| | Channel Shuffle | 0.1 |
| | Gaussian Noise | 0.1 |
| | Median Blur | 0.1 |
| | Sharpness | 0.1 |
| Geom. | Thin Plate Spline | 0.1 |
| | Crop (2×2) | 0.1 |
| | Random Erasing | 0.1 |
| Mix | CutMix V2 | 0.1 |

Unlike the conventional classification scenario where a single prediction $y^{(t)}$ is obtained, the combination of TTA and TTD techniques enables us to model a predictive distribution. This novel approach involves training the network as a typical neural network, but with slight modifications to the process.

$$y^{(t)} = \arg\max_k f_\theta^{(t)}(\tilde{x}), \quad \text{for } t = 1, \dots, T \tag{1}$$

$$\hat{y} = \text{mode}\left(\{y^{(1)}, y^{(2)}, \dots, y^{(T)}\}\right) \tag{2}$$

$$\text{Mean Confidence}(\hat{y}) = \frac{1}{T}\sum_{t=1}^{T} \mathbb{I}[y^{(t)} = \hat{y}] \tag{3}$$

Eqs (1) to (3) describe how we estimate prediction confidence using multiple stochastic forward passes through the model. In Eq (1), we apply random image augmentations and enable dropout during test time to generate diverse predictions $y^{(t)}$ for the same input image. This process is repeated $T$ times to simulate uncertainty due to both data variability and model limitations [36]. In Eq (2), we identify the most frequently predicted class across all $T$ iterations, known as the mode, which becomes the final predicted label $\hat{y}$. Eq (3) then calculates the confidence score as the proportion of predictions that agree with this final class. This gives a simple but effective estimate of how certain the model is in its prediction the higher the confidence, the more consistent the model's predictions were under different random conditions [43].

## Experimental setup

In Fig 1, our methodological approach is illustrated in detail. Initially, bounding boxes are created for 29,374 images to crop them, ensuring that only the entire body of the Diptera is visible while removing any background as seen in Fig 1(A). Bounding Boxes were first generated using a pre-trained object detection model from [25] and then manually corrected by two expert ecologists using the open-source tool LabelImg in a Python environment. This preprocessing step is crucial to focus the data set on the relevant features. Subsequently, three CNNs are trained using both the cropped and uncropped image datasets to assess the impact of background removal on classification performance as seen in Fig 1(B). Following this, we quantify the confidence of our predictions by approximating uncertainty estimates using TTA and TTD, with 100 Monte Carlo iterations providing robust statistical analysis. Finally, we compare the classification results, including confidence values, between the uncropped and cropped images, allowing a comprehensive evaluation of our methodological approach as seen in Fig 1(C).

The preprocessing pipeline for the images uses a z-score normalization for the images using the statistical parameters specific to the ImageNet dataset. This normalization ensures that the input images maintain consistency with the pre-training dataset, aligning their distribution with the expectations of the pre-trained model [41,46].

Moreover, z-score normalization serves as a pivotal data preprocessing step due to its ability to standardize the input data, mitigating the impact of varying scales and intensities within the image data [47]. This is particularly crucial when working with diverse datasets that may exhibit significant variations in illumination conditions or imaging equipment. This improves the adaptability of the models to the unique features present in the Diptera image dataset, ultimately fostering their ability to capture and learn meaningful patterns during the training process.

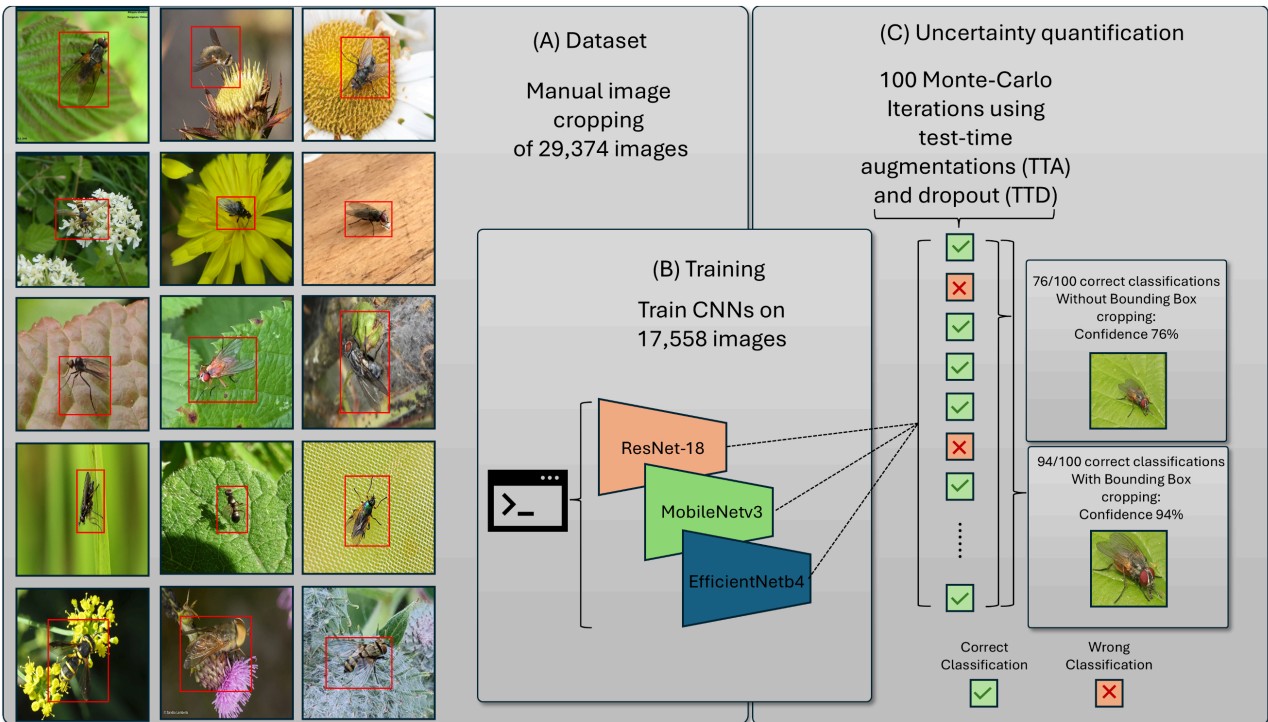

**Fig 1. Uncertainty quantification approach.** (A) Creation of bounding boxes around Diptera specimens to generate both cropped and uncropped versions of the dataset. (B) Training of CNNs using both image variants to evaluate the effect of cropping on classification performance. (C) Confidence estimation using test-time augmentation (TTA) and test-time dropout (TTD) with Monte Carlo sampling to assess the model confidence. While this figure shows an example for one Family, Fannidae, the uncertainty quantification was applied across all families in the dataset.

Each CNN undergoes training for a consistent duration of 100 epochs. To ensure a robust evaluation, we present all metrics and results based on the model saved with the best validation score. In this context, the optimal model is determined by achieving the lowest validation score for the cross-entropy loss, employing label smoothing. This approach guarantees that the reported results reflect CNN's performance at its peak during the training process, providing a comprehensive and accurate assessment of its capabilities.

In this study, we systematically evaluated the tradeoff between the per-epoch training speed of a CNN and the total time required to reach optimal validation performance. Specifically, we investigated how various training settings affect convergence and generalization. Our experiments used 8-bit RGB images resized to 640×640 pixels and normalized using z-score normalization based on ImageNet statistics. The models were trained for a maximum of 100 epochs using early stopping, with cross-entropy loss incorporating label smoothing (0.1) and the AdamW optimizer. A fixed learning rate of 0.001 was combined with a CosineAnnealing-WarmRestarts scheduler to allow for dynamic adjustment during training. By keeping these settings consistent, we aimed to strike a balance between fast per-epoch training and reliable generalization to unseen data, while minimizing the risk of overfitting.

In addition, this evaluation was conducted with an emphasis on energy efficiency and environmental considerations, reflecting the principles of Green AI [48]. Specifically, we incorporated resource usage measurements and computational cost analyzes into our training protocols to ensure that performance gains did not come at the expense of excessive energy consumption [15]. Taking into account both training speed and environmental impact, this

approach supports more responsible model development practices, aligning model optimization strategies with sustainability goals.

The code used to collect and process the data is publicly available on our GitHub repository. You can access both the data collection code and the data processing code here stark-t/PAI_diptera.

## Results

### Comparative analysis of CNNs

Table 4 compares three CNNs in terms of accuracy and efficiency, detailing their number of parameters, overall accuracy (OA), Kappa score, mean confidence, training epochs, time per epoch, test time for 100 Monte Carlo iterations, and whether the image was cropped to its bounding box. OA and Kappa are calculated as shown in Eqs (4) and (5), where the Kappa score incorporates the expected agreement by chance (Eq (6)) to provide a more robust evaluation for imbalanced datasets, mean confidence is calculated using Eq (3).

$$OA = \frac{TP + TN}{TP + TN + FP + FN} \tag{4}$$

$$\kappa = \frac{p_o - p_e}{1 - p_e} \tag{5}$$

where $p_o$ = Accuracy is the observed agreement, and $p_e$ is the expected agreement by chance, defined as:

$$p_e = \frac{(TP + FP)(TP + FN) + (FN + TN)(FP + TN)}{(TP + TN + FP + FN)^2} \tag{6}$$

Overall, we see very high accuracies in the high 80% range for all model architectures and images or image sections, and in some cases well over 90%. Basically, we can therefore state that the 15 families can be classified very well. The overall accuracy (OA) for MobileNetV3 Large using the original images is 88.78%, while ResNet-18 and EfficientNetB4 achieve 88.58% and 90.35%, respectively.

The number of parameters for each model correlates with these results. MobileNetV3 Large, with 5.4 million parameters, shows lower OA compared to more complex models. ResNet-18, having 11.3 million parameters, performs similar to MobileNetV3 Large but

**Table 4. Comparison of three CNNs in terms of accuracy and efficiency.** The table includes the number of parameters, overall accuracy (OA), Kappa score, mean confidence, training epochs, time per epoch, test time for 100 Monte Carlo iterations, and if the image was cropped to its bounding box

| Model | Parameters | OA | Kappa | Mean Confidence | Epochs | Epoch Time | Test Time | Bounding Box Crop |
|---|---|---|---|---|---|---|---|---|
| MobileNetV3 Large | 5.4m | 88.78% | .8887 | 78.49% | 47/100 | 00:05:16 | 01:53:27 | X |
| ResNet-18 | 11.3m | 88.58% | .8895 | 79.17% | 95/100 | 00:04:57 | 01:55:57 | X |
| EfficientNetB4 | 17.6m | 90.35% | .9044 | 79.27% | 46/100 | 00:11:34 | 02:22:32 | X |
| MobileNetV3 Large | 5.4m | 93.87% | .9401 | 85.85% | 94/100 | 00:05:23 | 01:46:30 | ✓ |
| ResNet-18 | 11.3m | 93.16% | .9384 | 83.98% | 54/100 | 00:04:46 | 01:49:16 | ✓ |
| EfficientNetB4 | 17.6m | 95.61% | .9590 | 87.38% | 70/100 | 00:11:23 | 02:29:24 | ✓ |

slightly better. EfficientNetB4, which has the highest number of parameters at 17.6 million, consistently shows superior OA in both sets of experiments.

When images are cropped to their bounding box, the results generally show a significant improvement. For MobileNetV3 Large, the OA increases from 88.78% to 93.87%, which is an improvement of approximately 5.09. ResNet-18's OA rises from 88.58% to 93.16%, an enhancement of about 4.58. EfficientNetB4 also benefits significantly, with its OA increasing from 90.35% to 95.61%, reflecting an improvement of approximately 5.26. In general, cropping images to their bounding box results in increased accuracies across the models, highlighting the importance of expert-defined bounding box image cropping in boosting model performance.

A similar pattern is observed in the Kappa scores, which provide a more robust measure by accounting for chance agreement. MobileNetV3 Large improves from a Kappa of 0.8887 (uncropped) to 0.9401 (cropped). ResNet-18 increases from 0.8895 to 0.9384, and EfficientNetB4 from 0.9044 to 0.9590. These high Kappa values further confirm that the classification performance is not only accurate but also consistent across the different models and image configurations.

Examining the mean confidence, MobileNetV3 Large has a mean confidence of 78.49% for the original images and 85.85% when cropping the images in its bounding box. ResNet-18 records 79.17% and 83.98%, while EfficientNetB4 demonstrates 79.27% and 87.38%, respectively.

Averaging across the three CNN architectures, the use of cropped images leads to a clear improvement in all performance metrics. The mean overall accuracy increases from 89.24% (uncropped) to 94.21% (cropped). Similarly, the mean Kappa score rises from 0.8942 to 0.9458, indicating improved agreement beyond chance. The mean confidence also shows a substantial boost, increasing from 78.98% to 85.74%. These results highlight that cropping images to their bounding boxes not only improves classification accuracy but also enhances model certainty and reliability.

## Class confusion between Diptera families with and without Bounding Box Cropping

Fig 2 compares the confusion matrices for the EfficientNetB4 model, highlighting the classification performance across 15 Diptera families under two different conditions: (a) images cropped to their bounding box and (b) images not cropped.

From the matrices, it is evident that cropping images to their bounding box significantly enhances the model's classification accuracy. This is reflected in higher true positive rates and reduced misclassification rates. For example, the Anthomyiidae family shows an accuracy of 89.9% with cropped images, compared to 86.6% with uncropped images. Similarly, for the Muscidae family, the accuracy is 92.4% when images are cropped, whereas it drops to 76.2% for uncropped images.

Moreover, the confusion within classes is noticeably higher when images are not cropped. This increased confusion is visible in the higher percentages of misclassification in several families. For example, in the Muscidae family, we observe that 7.2% of the samples are misclassified as Anthomyiidae, 4.6% as Calliphoridae and 3.5% as Tachinidae when the images are not cropped. In contrast, when the images are cropped, these misclassification rates drop significantly, showcasing more accurate predictions.

Similar patterns are observed in other families. For instance, the Sarcophagidae family, when images are uncropped, has misclassifications of 7.0% into Tachinidae and 3.7% into

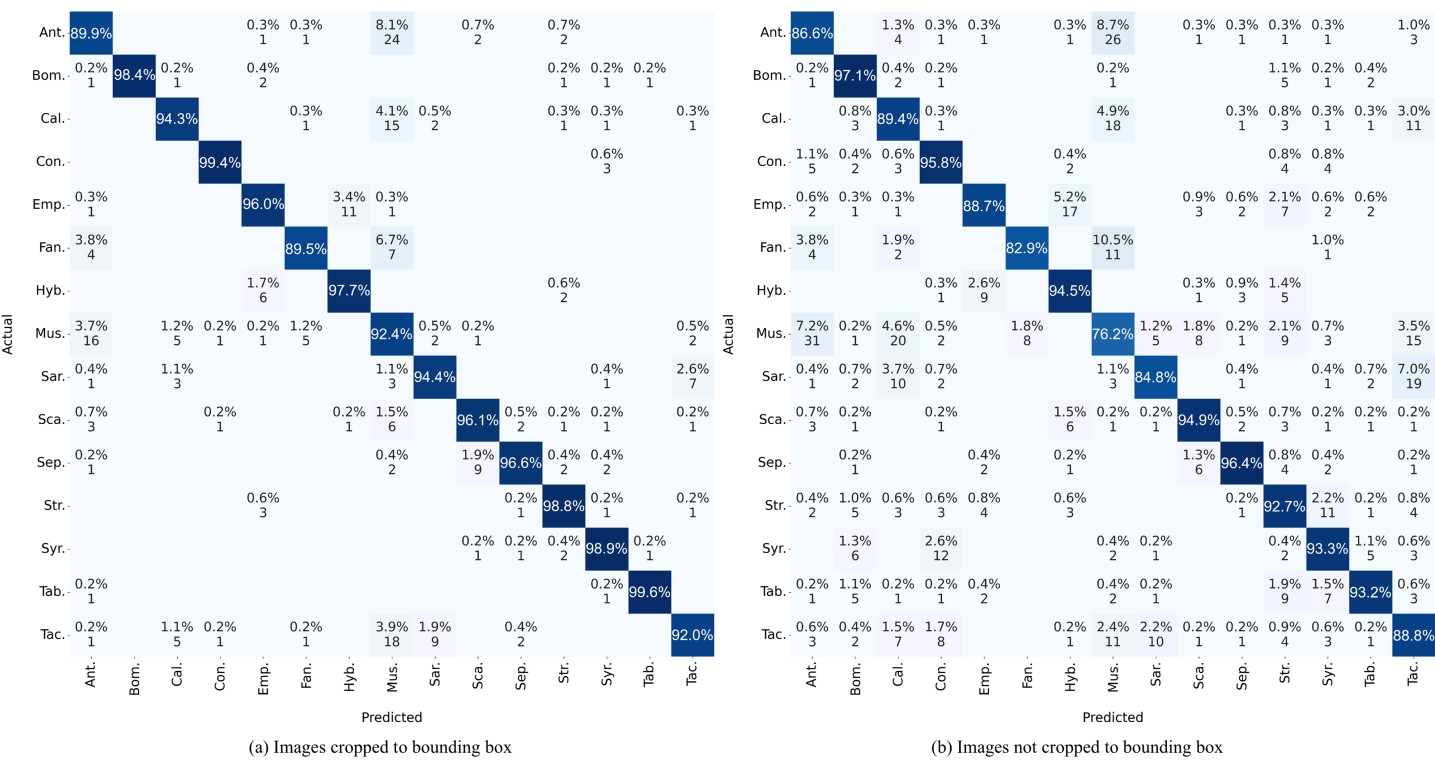

(a) Images cropped to bounding box          (b) Images not cropped to bounding box

**Fig 2. Confusion matrix comparing EfficientNetb4 results: (a) training with images cropped to the bounding box and (b) training with uncropped images.**

Calliphoridae. However, when cropped, the misclassification rates are significantly reduced, leading to more precise classification.

Overall, the results indicate that preprocessing images by cropping them to their bounding box substantially reduces confusion and improves the EfficientNetB4 model's accuracy and reliability in classifying the 15 Diptera families.

## The uncertainty within the Diptera families

Fig 3 presents a boxplot illustrating the confidence values of the EfficientNetB4 model for all 15 Diptera families, comparing the results for the original images and the images cropped to their bounding box. This graph highlights the confidence levels for the correctly predicted images.

We sorted the families by the magnitude of improvement in their confidence scores, that is, the difference between the cropped image confidence and the original image confidence. Families with an increase of 6% or more are categorized as having large improvements and are listed first in descending order, while those with less than 6% improvement are grouped as small improvements, also in descending order.

In the group of large improvements, Stratiomyidae experiences the largest increase, with confidence increasing from 84% to 94% (a 10% improvement). Muscidae and Fanniidae follow with significant increases of 9% each, with Muscidae improving from 80% to 89% and Fanniidae from 78% to 87%. Both Calliphoridae and Sarcophagidae show a 6% improvement, with Calliphoridae moving from 82% to 88% and Sarcophagidae from 81% to 87%.

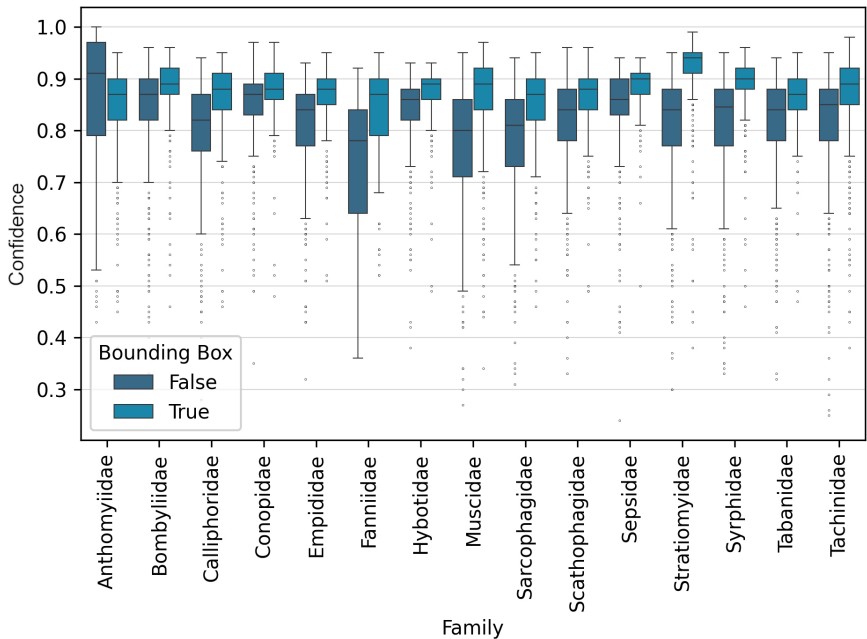

**Fig 3. Confidence values for EfficientNetB4 for all 15 Diptera families for correct classification results with and without cropping its Bounding Boxes.**

In the small improvements group, Syrphidae's confidence increases from 85% to 90% (a 5% improvement). Sepsidae improve from 86% to 90% (a 4% increase), as do Empididae and Scathophagidae, each increasing from 84% to 88% (a 4% improvement). Tachinidae also shows a 4% increase, moving from 85% to 89%. Hybotidae experiences a 3% improvement, going from 86% to 89%, and Tabanidae likewise improves by 3%, rising from 84% to 87%. Bombyliidae sees a modest 2% increase from 87% to 89%, while Conopidae remains unchanged at 88%.

Anthomyiidae is the only family that shows a decrease in median confidence, falling from 91% to 87%.

## Visual results for all 15 families

The visual results of the analysis of the 15 Diptera families highlight significant differences between the two training approaches using the best performing model EfficientNetB4. Specifically, we compared the performance when the model was trained on images cropped to the Diptera bounding box versus images that were not cropped. Fig 4 presents two images from the test dataset for each Diptera family, highlighting both the highest and lowest confidence differences between the images cropped to its bounding box and the uncropped ones. Moreover, these images were automatically selected rather than hand-picked, ensuring an unbiased and representative illustration of the model's performance. Generally, as illustrated in Fig 3, the model trained on cropped images consistently achieves better results.

## Discussion

Fig 4 presents a comprehensive comparison of models trained with and without cropping to the Diptera's bounding box across the 15 families. For enhanced visual clarity, the species

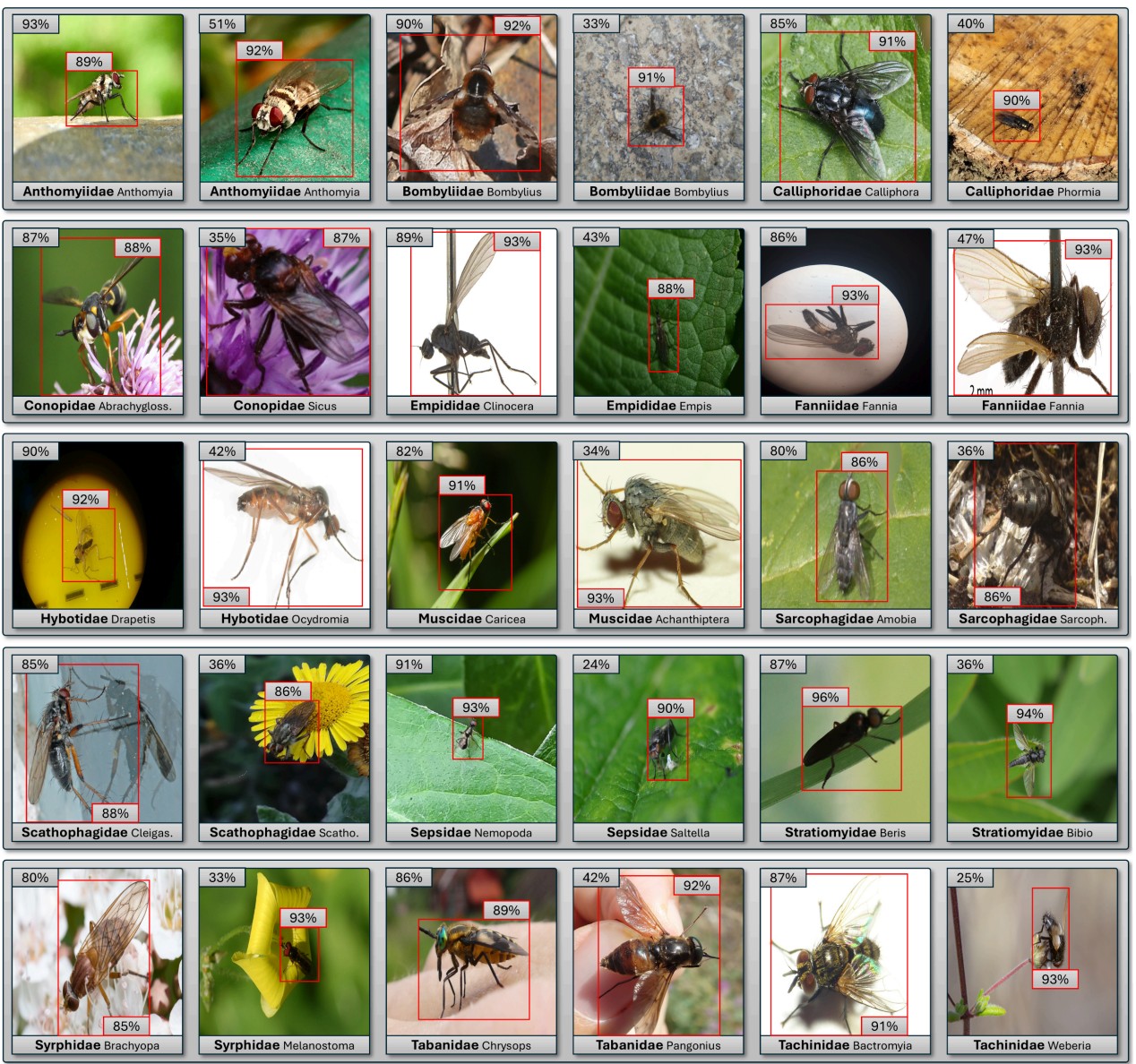

**Fig 4. Comparison of EfficientNetB4 confidence values on 15 Diptera families.** The figure highlights confidence differences between models trained on images cropped to the Diptera's bounding box (red bounding box) versus those using (full) images . For each Diptera family, two examples are shown: one representing the smallest difference and one representing the largest difference in confidence values of the test dataset.

for each family are displayed solely in this figure. Note that this information was not used in the classification process but is provided only to offer additional context for interpreting the results.

Creating bounding boxes manually is an extremely time-consuming process. For the 29,374 images used in this study, bounding boxes had to be created by experts, following a consistent workflow to ensure that each specimen is cropped uniformly. This task requires not only significant time but also specialized expertise to accurately delineate the objects of interest. While the manual annotation process is undeniably labor-intensive, our results highlight

a critical trade-off: the substantial effort required for cropping is justified by notable improvements in overall accuracy and mean confidence, suggesting that such manual intervention can significantly enhance model performance. Also semi-automated labeling tools, could further support and streamline this process. It would be very interesting to explore the use of object detection algorithms to automatically crop images similar to the methods used in [49] and [25]. Moreover, comparing the performance of models trained on manually cropped images with those cropped automatically could provide valuable insights into both the feasibility and efficiency of automated cropping.

In most cases, cropping improves performance by focusing the model on the relevant features of the Diptera, thereby reducing background noise. This improvement is likely due to the model's increased focus on Diptera features, which minimizes background noise and facilitates more effective feature extraction, which ultimately leads to greater overall precision [50]. For example, within the Calliphoridae family, the Phormia specimen occupies only a small fraction of the full image, resulting in a dramatic increase in the predicted confidence, from 40% without cropping to 90% with cropping. In contrast, for Calliphoridae Calliphora, where the diptera fills almost the entire image, the confidence values are much closer (91% with cropping vs. 85% without cropping). This trend of improved performance with cropping is evident in several families.

However, there are some notable outliers. In cases such as Conopidae Sicus, the second example of Fanniidae Fannia, Hybotidae Ocydromia, Muscidae Achanthiptera, and Tabanidae Pangonius, the bounding box is only slightly smaller than the original image, yet the differences between the cropped and non-cropped models are pronounced. In many of these cases, the uniform laboratory environment in the background could significantly influence the model's training. In addition, challenging scenarios are observed in Sarcophagidae Sarcophaga, where the Diptera closely resembles the background, and in Scathophagidae Cleigastra, where the specimen is imaged on a highly reflective surface, both cases that test the model's adaptability under varying conditions.

## Challenges in Diptera classification

Distinguishing between these 15 Diptera families from images is a challenging task, because many families share numerous morphological features, and certain distinguishing characteristics might not be visible in images. Fig 5 illustrates these challenges by comparing the results of three morphologically similar Diptera families. In such scenarios, even a trained expert struggles to confidently separate closely related groups by manual-visual interpretation, leading to potential misclassification when relying on visual cues alone. Despite these challenges, our models demonstrate robust performance. However, overfitting remains a concern in many deep learning algorithms. Models may exploit subtle cues that are not directly linked to the underlying taxonomy, and dataset sampling biases can also contribute to overfitting. This possibility underscores the critical need for our uncertainty estimation approach, which integrates extensive image augmentations and dropout to mitigate these effects.

By incorporating 100 Monte Carlo iterations, we gain a quantitative window into the confidence scores of the model, enabling us to confirm that the impressive precision is neither coincidental nor artificially inflated. Instead, these analyzes demonstrate that the model's capabilities extend beyond pattern matching and that it genuinely understands the underlying structure of the data. In fact, this examination reveals that the model can consistently identify subtle class distinctions, even in fuzzy feature spaces where related classes share overlapping traits. The combination of strong performance and transparent uncertainty quantification offers both confidence and credibility.

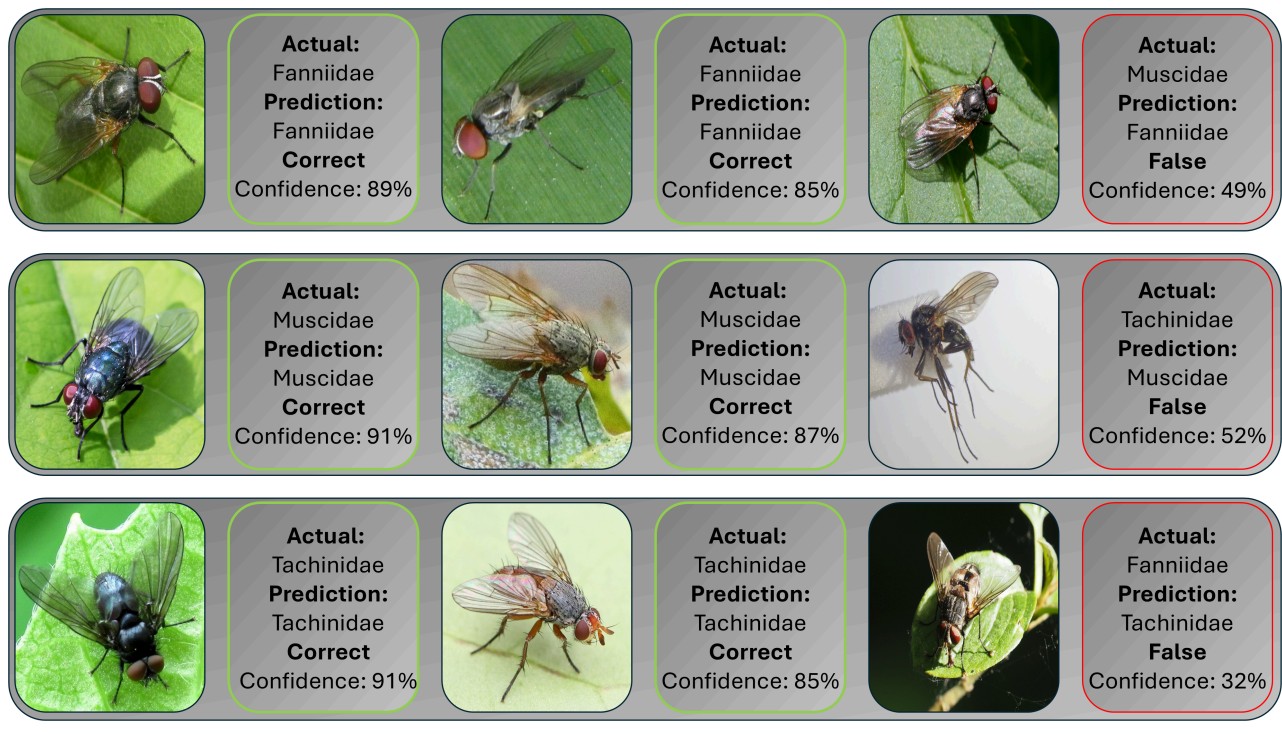

**Fig 5. Confidence values for three Diptera families with morphological similarities.** Predicted and actual labels for selected examples classified by EfficientNetB4 using images cropped to their bounding boxes. The model's predicted class, corresponding ground truth label, and prediction confidence are shown for each image.

As observed in [49], cropping images to the insect's bounding box has a significant effect on enhancing model performance. This improvement is expected since the models do not need to process excessive background information, which can be distracting. Distracting backgrounds can have a substantial impact on model accuracy.

The results of our study highlight the effectiveness of different CNN architectures in classifying Diptera families. Although earlier generation models such as ResNet-18 provide acceptable accuracy, more recent models, such as EfficientNet-B4, demonstrate better performance, offering higher confidence and fewer misclassifications. For scenarios where speed and portability are critical, lightweight models such as MobileNetv3 deliver strong results, enabling on-device classification without significantly compromising accuracy. In this way, we see a clear progression in model capabilities, from established architectures to cutting-edge networks, each offering specific advantages that can be leveraged according to the demands of the task.

## Confusion between Diptera families

Four families were most often confused with each other: Anthomyiidae, Fanniidae, Muscidae, and Tachinidae, all of which belong to the same superfamily, Muscoidea. Their morphological similarities often complicate accurate identification. In particular, wing venation patterns serve as a critical feature to distinguish these families, but it is challenging to capture clear and properly angled images to visualize the entire wing. Despite these difficulties, it is rewarding to see that the models perform as well as they did.

To illustrate this performance, we present confidence values for a subset of these morphologically similar Diptera families seen in Fig 5. Family names in bold indicate the true labels,

while underlined family names represent the model's predictions. These confidence scores, obtained using an EfficientNetB4 model with bounding box cropped images, underscore the model's capability to differentiate among closely related taxa.

From the confusion matrix in Fig 2a, we observe that of the 105 Fannidae images in our test dataset, seven are misclassified as Muscidae. One such misclassification is shown in Fig 5, where EfficientNet exhibits low confidence (49%) in its classification. In contrast, correctly classified Fannidae images have higher confidence scores of 85% and 89%. Overall, the mean confidence for Fannidae predictions is 0.78.

For the Muscidae family, five of 433 test images are misclassified as Fannidae and two as Tachinidae. A misclassification of Tachinidae is depicted in Fig 5, with a low confidence of 52%. Correct classifications for Muscidae have higher confidence scores of 87% and 91%. The mean confidence for Muscidae predictions is slightly higher at 0.89. This difference in mean confidence could be partially attributed to class imbalance, as the Fannidae family has significantly fewer training examples than Muscidae, potentially limiting the model's ability to generalize confidently.

In the Tachinidae family, one out of 465 test images is misclassified as Fannidae, and 18 as Muscidae. The single misclassification as Fannidae is shown in Fig 5, with a very low confidence score of 32%. The correct classifications of Tachinidae have confidence scores of 85% and 91%. Thus, in general, for the purposes of ecological studies, when confidence scores are 85% or higher, it is reasonable to identify the fly in the image to the family taxonomic level. However, when the confidence level is less than 50%, it is advisable to keep the identification at the order level (Diptera).

The few misclassifications observed are informative as well. They highlight the areas where the model can be further refined, and they provide insight into the limits of current image recognition technology in dealing with highly similar morphological features. This approach is a significant step forward in the field of automated species identification and has the potential to contribute meaningfully to various scientific and practical applications.

The next step in this line of research is to delve deeper into the taxonomic hierarchy. The ability to classify pollinating insects at the species levels automatically from images would be highly beneficial to ecological monitoring [26]. This progression will test the limits of our current methodologies and may necessitate the development of new approaches. Although our existing models have proven effective at higher taxonomic levels, the increased specificity required at the species levels could demand more sophisticated architectures, such as transformers, as well as larger and more well sampled datasets. We anticipate that this new challenge will be complex, yet we remain optimistic about our ability to overcome potential obstacles. Adapting and refining our methods to achieve fine-scale classification will not only advance our understanding of pollinator diversity but also contribute to broader ecological and conservation efforts.

## Conclusion

In conclusion, our study demonstrates that automated pollinator monitoring using CNNs can overcome many of the limitations of traditional methods, offering a cost-effective and efficient alternative resulting in high classification accuracies. By focusing on 15 European pollinating fly families, a group that has traditionally been challenging due to subtle morphological differences, we showed that CNN architectures such as ResNet18, MobileNetV3, and especially EfficientNetB4, can achieve high classification accuracy (from 88.58% to 95.61%). A key finding was that cropping images to the Diptera's bounding boxes not only improved accuracy but also enhanced prediction certainty, effectively reducing misclassifications among families.

Visual analyses further corroborate these results. In most cases, especially when the fly occupies only a small portion of the image, cropping significantly increased the predicted confidence, thereby reinforcing the benefits of focusing on the relevant features. Although some outliers were observed, possibly due to uniform lab environments or challenging imaging conditions, the overall trend supports the use of bounding-box cropping as a robust method for improving model performance.

This work not only advances the field of automated pollinator monitoring, but also provides a foundation for future applications in ecological research and practical conservation efforts, ensuring that critical pollination services are better understood and protected.

## Author contributions

**Conceptualization:** Thomas Stark, Michael Wurm, Valentin Stefan, Hannes Taubenböck, Tiffany M. Knight.

**Data curation:** Thomas Stark, Valentin Stefan, Felicitas Wolf, Tiffany M. Knight.

**Formal analysis:** Thomas Stark, Michael Wurm.

**Funding acquisition:** Hannes Taubenböck.

**Investigation:** Thomas Stark, Michael Wurm, Valentin Stefan, Tiffany M. Knight.

**Methodology:** Thomas Stark.

**Project administration:** Thomas Stark, Michael Wurm, Hannes Taubenböck, Tiffany M. Knight.

**Resources:** Thomas Stark, Valentin Stefan, Tiffany M. Knight.

**Software:** Thomas Stark, Valentin Stefan, Felicitas Wolf.

**Supervision:** Michael Wurm, Hannes Taubenböck, Tiffany M. Knight.

**Validation:** Thomas Stark.

**Visualization:** Thomas Stark.

**Writing – original draft:** Thomas Stark, Tiffany M. Knight.

**Writing – review & editing:** Michael Wurm, Valentin Stefan, Felicitas Wolf, Hannes Taubenböck.

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
