## [Decision Letter · Decision Letter 0]

2 Jun 2025

PONE-D-25-20622Utilizing CNNs for classification and uncertainty quantification for 15 families of European fly pollinatorsPLOS ONE

Dear Dr. Stark,

Thank you for submitting your manuscript to PLOS ONE. After careful consideration, we feel that it has merit but does not fully meet PLOS ONE’s publication criteria as it currently stands. Therefore, we invite you to submit a revised version of the manuscript that addresses the points raised during the review process.

We look forward to receiving your revised manuscript.

Kind regards,

Ramzi Mansour

Academic Editor

PLOS ONE

Reviewers' comments:

Reviewer's Responses to Questions

**Comments to the Author**

1. Is the manuscript technically sound, and do the data support the conclusions?

Reviewer #1: Yes

Reviewer #2: Partly

2. Has the statistical analysis been performed appropriately and rigorously? 

Reviewer #1: Yes

Reviewer #2: I Don't Know

3. Have the authors made all data underlying the findings in their manuscript fully available?

Reviewer #1: Yes

Reviewer #2: Yes

4. Is the manuscript presented in an intelligible fashion and written in standard English?

Reviewer #1: Yes

Reviewer #2: Yes

5. Review Comments to the Author

Reviewer #1: 

This is a very well written and presented manuscript; discussion and conclusions are well thought through, and relevant literature used. It will be of interest to many others regarding methods and overall conclusions.

Minor issues

• Line 168 – how did the bounding box but also the preprocessing get done, what software environment?

• Fig3 misleading legend, should say "With" and "Without" (rather than False/True), the figure can be interpreted differently with the words False/True

• Fig 3 caption – spelling “boundinx", should be Bounding

• Abstract states "Our dataset comprises a wide range of morphological and phylogenetic features, such as wing venation patterns and wing shapes" but this is misleading; the images used were general habitus images of the body. Yes "features of the wings" are part of this general body but they are not the only feature. The statement in the abstract implies wing features are the central part of the images.

• Early discussion (lines 305 +) regarding the time to make bounding boxes; unsure if the authors know of Roboflow (www.roboflow.com) - it has some semi automated features and is very easy to use.

One thing the authors could examine in any future work is the ranges and overlap of confidence scores of those images that are scored correctly vs incorrectly. There is some discussion of this on lines around 380-390. This connects with "trustworthiness" that the authors mention in the introduction. It is an issue that we are facing in our lab group - where is there overlap in scores between right and wrong, and so it is difficult for the user to be confident of the score.

Reviewer #2: 

General comments

I enjoyed reading this study and there were some interesting methodology and results however some information was lacking. Further details are required in the materials and methods for transparency and the figure captions need fuller explanations so that if viewed independently the reader understands them.

A key aspect of this study are the uncertainty quantification and mean confidence calculations; these provide a potentially useful metric for researchers using machine learning classifiers however some methodological details are missing. There is also some imbalance in the number of images for each of the families and I would be interested to see the f1 scores to determine whether this has impacted CNN performance.

If these issues, further detailed in the comments below were addressed I would recommend this study for publication.

Abstract

I would like more information about the methodology here and some more results – e.g. what method did you use to determine accuracy and prediction confidence? What was the improvement in accuracy using bounding boxes?

Introduction

Line 12: A definition of a CNN would be useful here.

Line 28: Further expansion of this paragraph would help to highlight the rationale for your study

Line 36: Clarification of uncertainty estimation in relation to previous works and how it appliesto this study would be beneficial in this paragraph.

Line 58: a brief description of your uncertainty quantification methods in this paragraph would emphasise the novelty of your approach

Materials and Methods

General - Further information about Diptera and their identification features would be interesting, particularly examples of wing venation. Including the formulae for the mean confidence score would provide further clarity.

Line 79: I think this paragraph would be better placed in the introduction

Line 147: Which image augmentations were used and in which proportions?

Line 154: What percentage of dropout did you use?

Line 160/161: Did you use augmented data when you trained these models?

Figure 1: More detail in the caption is needed – are the confidence figures for all species?

Line 197: This paragraph is a little ambiguous, which configurations did you investigate? Learning rate, batch size etc?

Experimental set up: Information regarding the performance metrics and how they are calculated is missing. The Kappa score is calculated but not explained.

Results

Line 217: I think you are understating your results a little here, all your models achieved OAs in excess of 88% and mean confidence of more than 78%. The Kappa score could also be mentioned here.

Line 232 and 235: You could add the mean improvement of OA and mean confidence across all CNNs for further clarity.

Figure 2: I would include the percentages only in the confusion matrices to make the figure more readable

The explanation of figure 5 is missing in this section.

Discussion

Line 309: The sentence “The labor-intensive nature of this manual approach underscores the potential benefits of automation” seems to contradict the contribution of your work – I think a more interesting issue is the trade-off between the labour required to crop the images and the improvement in OA and mean confidence scores.

Figure 5: This figure is somewhat confusing – are the green boxes examples of two different species from the same family that have been correctly predicted? Further information in the legend would clarify this.

Confusion between Diptera families: In the second and third paragraphs, the two families you are comparing Fanniidae and Muscidae have 521 and 2163 images respectively, what impact does this class imbalance between have on their OA and mean confidence?

6. PLOS authors have the option to publish the peer review history of their article (what does this mean?). If published, this will include your full peer review and any attached files.

Reviewer #1: **Yes: **Darren Ward

Reviewer #2: No

---

## [Author Response · Author response to Decision Letter 1]

23 Jul 2025

Response to the reviewers

Title: Utilizing CNNs for classification and uncertainty quantification for

15 families of European fly pollinators

Manuscript Number: PONE-D-25-20622

Dear Irene Nathalie Fernandez Tolentino,

Dear Editors,

Thank you very much for considering our manuscript for publication PLOS ONE subject to the

requested revisions based on reviewers’ comments. Please find below a point-by-point answer to

the comments (bold, cursive and bullet points) in blue and new text which was added to the

main manuscript in red.

We acknowledge the valuable reviewers’ comments and hope that the revised manuscript meets

all specifications with respect to the changes made.

Sincerely,

Thomas Stark

Reviewer #1:

This is a very well written and presented manuscript; discussion and conclusions are well

thought through, and relevant literature used. It will be of interest to many others regarding

methods and overall conclusions.

• (Line 168) How did the bounding box but also the preprocessing get done, what software

environment?

Thank you for bringing this to our attention we added more detail of making the bounding boxes

and expanded the preprocessing process in more detail.

The bounding box generation was conducted in a two-step process. First, we used a pre-trained

object detection model based on the approach described in [23] to automatically generate

initial bounding boxes around the Diptera specimens. In a second step, these boxes were

manually refined by two expert ecologists to ensure high annotation accuracy. This manual

correction was performed for all ~30,000 images using the open-source tool LabelImg within a

Python-based environment. This combination of automated detection and expert validation

ensured both efficiency and high-quality ground truth for subsequent model training.

(L.213-218) In Figure 1 our methodological approach is illustrated in detail. Initially, bounding

boxes are created for 29,374 images to crop them, ensuring that only the entire body of the

Diptera is visible while removing any background as seen in Figure 1(A). Bounding Boxes were

first generated using a pre-trained object detection model from [23] and then manually

corrected by two expert ecologists using the open-source tool LabelImg in a Python

environment.

We are not entirely sure whether the preprocessing refers to image preprocessing or data

selection preprocessing. Nevertheless, for image preprocessing, we applied z-score

normalization using the RGB mean and standard deviation values from ImageNet, which we also

used for pretraining our three CNN models. This normalization helps standardize pixel

distributions across images and is commonly used in deep learning pipelines to improve model

convergence and performance [32,37,38]

Regarding dataset curation, this was a more involved step due to the high variability of GBIF

images. We invested considerable effort into selecting images that depict adult Diptera on

flowers or in natural environments, while excluding larval stages, museum specimens, or highly

artificial backgrounds. While GBIF image tags were partially helpful for filtering, most of this

refinement took place during the bounding box creation and correction process, where we

manually removed unsuitable samples to ensure a consistent and high-quality dataset.

• Fig3 misleading legend, should say "With" and "Without" (rather than False/True), the

figure can be interpreted differently with the words False/True

Thank you for highlighting this typo, we fixed it in the revised manuscript.

• Fig 3 caption – spelling “boundinx", should be Bounding

Thank you again for highlighting this typo, we fixed it in the revised manuscript.

Fig 3. Confidence values for EfficientNetB4 for all 15 Diptera families for correct classification

results with and without cropping its Bounding Boxes.

• The abstract states "Our dataset comprises a wide range of morphological and

phylogenetic features, such as wing venation patterns and wing shapes" but this is

misleading; the images used were general habitus images of the body. Yes "features of

the wings" are part of this general body but they are not the only feature. The statement

in the abstract implies wing features are the central part of the images.

Thank you for highlighting this point. We agree that the original phrasing may have been too

strong and will soften the wording in the abstract accordingly.

Pollination is essential for maintaining biodiversity and ensuring food security, and in Europe it

is primarily mediated by four insect orders (Coleoptera, Diptera, Hymenoptera, Lepidoptera).

However, traditional monitoring methods are costly and time consuming. Although recent

automation efforts have focused on butterflies and bees, flies, a diverse and ecologically

important group of pollinators, have received comparatively little attention, likely due to the

challenges posed by their subtle morphological differences. In this study, we investigate the

application of Convolutional Neural Networks (CNNs) for classifying 15 European pollinating fly

families and quantifying the associated classification uncertainty. In curating our dataset, we

ensured that the images of Diptera captured diverse visual characteristics relevant for

classification, including wing morphology and general body habitus. We evaluated the

performance of three CNNs, ResNet18, MobileNetV3, and EfficientNetB4 and estimated the

prediction confidence using Monte Carlo methods, combining test-time augmentation and

dropout to approximate both aleatoric and epistemic uncertainty. We demonstrate the

effectiveness of these models in accurately distinguishing fly families. We achieved an overall

accuracy of up to 95.61%, with a mean relative increase in accuracy of 5.58% when comparing

uncropped to cropped images. Furthermore, cropping images to the Diptera bounding boxes not

only improved classification performance across all models but also increased mean prediction

confidence by 8.56%, effectively reducing misclassifications among families. This approach

represents a significant advance in automated pollinator monitoring and has promising

implications for both scientific research and practical applications.

• Early discussion (lines 305 +) regarding the time to make bounding boxes; unsure if the

authors know of Roboflow (www.roboflow.com) - it has some semi automated features

and is very easy to use.

Thank you for this helpful suggestion. We were only loosely aware of Roboflow but had not used

it in our workflow. Its semi-automated features indeed seem very helpful, and we now mention

in the discussion that such tools could support more efficient annotation in future work.

(L.375-388) Creating bounding boxes manually is an extremely time-consuming process. For the

29,374 images used in this study, bounding boxes had to be created by experts, following a

consistent workflow to ensure that each specimen is cropped uniformly. This task requires not

only significant time but also specialized expertise to accurately delineate the objects of

interest. The labor-intensive nature of this manual approach underscores the potential benefits

of automation. Also semi-automated labeling tools, could further support and streamline this

process. It would be very interesting to explore the use of object detection algorithms to

automatically crop images similar to the methods used in [49] and [25].

• One thing the authors could examine in any future work is the ranges and overlap of

confidence scores of those images that are scored correctly vs incorrectly. There is

some discussion of this on lines around 380-390. This connects with "trustworthiness"

that the authors mention in the introduction. It is an issue that we are facing in our lab

group - where is there overlap in scores between right and wrong, and so it is difficult for

the user to be confident of the score.

Thank you for this very interesting suggestion. We have also noticed similar issues in our results

and agree that analyzing the overlap in confidence scores between correctly and incorrectly

classified images is a promising direction. This is definitely a topic we will look into more closely

in future work, especially now knowing that other groups are encountering a similar challenge.

Reviewer #2:

I enjoyed reading this study and there were some interesting methodology and results however

some information was lacking. Further details are required in the materials and methods for

transparency and the figure captions need fuller explanations so that if viewed independently

the reader understands them. A key aspect of this study are the uncertainty quantification and

mean confidence calculations; these provide a potentially useful metric for researchers using

machine learning classifiers however some methodological details are missing.

• There is also some imbalance in the number of images for each of the families and I

would be interested to see the f1 scores to determine whether this has impacted CNN

performance.

Thank you for your valuable observation regarding potential class imbalance and the relevance

of F1 scores. While we did observe some variation in the number of images per fly family, we do

not consider the imbalance to be substantial enough to significantly skew model performance.

To evaluate this further, we did compute the macro-averaged F1 scores alongside other metrics.

However, we chose to focus on overall accuracy and Cohen’s kappa in the manuscript to

maintain clarity and avoid redundancy, as the F1 scores were very similar to these primary

metrics.

Nevertheless, we appreciate your suggestion and have now included a table comparing

accuracy, kappa, and F1 score per model and variant. As shown in the table below, the F1

scores follow the same trends as the accuracy and kappa metrics. For example, the EfficientNet

model with bounding box cropping (BB) achieved the highest values across all three metrics

(Accuracy: 95.61%, Kappa: 0.9590, F1: 0.9622), while the noBB variants consistently performed

lower. These results support our interpretation that minor class imbalance did not substantially

impact model performance.

Model Variant Accuracy Kappa F1

MobileNet noBB 88.78% 0.8887 0.8963

ResNet noBB 88.58% 0.8895 0.8973

EfficientNet noBB 90.35% 0.9044 0.9110

MobileNet BB 93.87% 0.9401 0.9444

ResNet BB 93.16% 0.9384 0.9426

EfficientNet BB 95.61% 0.9590 0.9622

• Abstract: I would like more information about the methodology here and some more

results – e.g. what method did you use to determine accuracy and prediction

confidence? What was the improvement in accuracy using bounding boxes?

Thank you for highlighting this important point. We have revised the abstract to include more

specific details regarding our methodology and key results. In particular, we now explicitly state

that overall accuracy (OA) was used as the main performance metric, and that prediction

confidence and uncertainty were quantified using Monte Carlo methods, combining test-time

augmentation and dropout to estimate both aleatoric and epistemic uncertainty.

Additionally, we clarified the performance gains achieved through image cropping. Specifically,

we now report that EfficientNetB4 reached an overall accuracy of 95.61%, representing a

relative improvement of approximately 5.82% over the uncropped version. We also mention

that, on average, cropping images to the bounding boxes of Diptera resulted in a 5.58% increase

in classification accuracy and an 8.56% increase in prediction confidence across all models.

These additions help emphasize the benefit of expert-defined cropping in boosting model

reliability.

We hope these clarifications improve the completeness and transparency of the abstract.

Pollination is essential for maintaining biodiversity and ensuring food security, and in Europe it

is primarily mediated by four insect orders (Coleoptera, Diptera, Hymenoptera, Lepidoptera).

However, traditional monitoring methods are costly and time consuming. Although recent

automation efforts have focused on butterflies and bees, flies, a diverse and ecologically

important group of pollinators, have received comparatively little attention, likely due to the

challenges posed by their subtle morphological differences. In this study, we investigate the

application of Convolutional Neural Networks (CNNs) for classifying 15 European pollinating fly

families and quantifying the associated classification uncertainty. In curating our dataset, we

ensured that the images of Diptera captured diverse visual characteristics relevant for

classification, including wing morphology and general body habitus. We evaluated the

performance of three CNNs, ResNet18, MobileNetV3, and EfficientNetB4 and estimated the

prediction confidence using Monte Carlo methods, combining test-time augmentation and

dropout to approximate both aleatoric and epistemic uncertainty. We demonstrate the

effectiveness of these models in accurately distinguishing fly families. We achieved an overall

accuracy of up to 95.61%, with a mean relative increase in accuracy of 5.58% when comparing

uncropped to cropped images. Furthermore, cropping images to the Diptera bounding boxes not

only improved classification performance across all models but also increased mean prediction

confidence by 8.56%, effectively reducing misclassifications among families. This approach

represents a significant advance in automated pollinator monitoring and has promising

implications for both scientific research and practical applications.

Introduction

• Line 12: A definition of a CNN would be useful here.

We have added a brief definition of Convolutional Neural Networks (CNNs) directly after their

first mention in the text to improve clarity for readers unfamiliar with the concept. Additionally,

we now include two standard references to foundational CNN [11, 12] to support this

explanation.

(L.10-15) The limited availability of images for flies, compared to more charismatic taxa of

pollinators, makes it difficult to create a comprehensive dataset, which is essential for

successfully training Convolutional Neural Networks (CNNs) [9, 10]. CNNs are a class of deep

learning models particularly well-suited for image recognition tasks due to their ability to

automatically learn hierarchical spatial features from raw pixels [11, 12].

• Line 28: Further expansion of this paragraph would help to highlight the rationale for your

study

Thank you very much for highlighting the importance of clearly positioning our study within the

broader context of deep learning applications for arthropod classification. In response, we have

revised and expanded the background section to better contextualize our work and explain its

rationale. Specifically, we now provide a more detailed overview of how deep learning has been

applied to various arthropod groups, citing several recent studies across different taxa. We also

added a dedicated paragraph emphasizing that pollinating flies—despite being the world’s

second most important group of pollinators after bees—have received comparatively little

attention in this research area. To support this point, we included several key references

[6,27,28,29] demonstrating the ecological and functional relevance of Diptera, especially in

regions or conditions where bees are less prevalent. These additions strengthen the motivation

for our study and highlight the gap our work aims to address: the need for robust deep learning

models capable of identifying pollinating flies, at least to the family level, for comprehensive

biodiversity monitoring and conservation efforts.

(L.32-41) However, pollinating flies have not yet been systematically addressed in this line of

research, despite being the world’s second most important and abundant pollinators after bees

[6,27]. Flies, particularly syrphids and other Diptera, make vital contributions to both natural

ecosystems an

---

## [Decision Letter · Decision Letter 1]

14 Aug 2025

Utilizing CNNs for classification and uncertainty quantification for 15 families of European fly pollinators

PONE-D-25-20622R1

Dear Dr. Stark,

We’re pleased to inform you that your manuscript has been judged scientifically suitable for publication and will be formally accepted for publication once it meets all outstanding technical requirements.

Kind regards,

Ramzi Mansour

Academic Editor

PLOS ONE

Additional Editor Comments (optional):

Reviewers' comments:

Reviewer's Responses to Questions

**Comments to the Author**

1. If the authors have adequately addressed your comments raised in a previous round of review and you feel that this manuscript is now acceptable for publication, you may indicate that here to bypass the “Comments to the Author” section, enter your conflict of interest statement in the “Confidential to Editor” section, and submit your "Accept" recommendation.

Reviewer #1: All comments have been addressed

Reviewer #2: All comments have been addressed

2. Is the manuscript technically sound, and do the data support the conclusions?

Reviewer #1: Yes

Reviewer #2: Yes

3. Has the statistical analysis been performed appropriately and rigorously? 

Reviewer #1: Yes

Reviewer #2: Yes

4. Have the authors made all data underlying the findings in their manuscript fully available?

Reviewer #1: Yes

Reviewer #2: Yes

5. Is the manuscript presented in an intelligible fashion and written in standard English?

Reviewer #1: Yes

Reviewer #2: Yes

6. Review Comments to the Author

Reviewer #1: (No Response)

Reviewer #2: I would like to thank the author for addressing my comments comprehensively and appreciate the improvements made to your article.

7. PLOS authors have the option to publish the peer review history of their article (what does this mean?). If published, this will include your full peer review and any attached files.

Reviewer #1: No

Reviewer #2: No

---

## [Editor Report · Acceptance letter]

PONE-D-25-20622R1

PLOS ONE

Dear Dr. Stark,

I'm pleased to inform you that your manuscript has been deemed suitable for publication in PLOS ONE. Congratulations! Your manuscript is now being handed over to our production team.

Kind regards,

on behalf of

Dr. Ramzi Mansour

Academic Editor

PLOS ONE